# Current Status and Trends of Minimally Invasive Gastrectomy in Korea

**DOI:** 10.3390/medicina57111195

**Published:** 2021-11-03

**Authors:** Shin-Hoo Park, Jong-Min Kim, Sung-Soo Park

**Affiliations:** 1Division of Foregut Surgery, Department of Surgery, Korea University College of Medicine, Goryeodae-ro 73, Seongbuk-gu, Seoul 02841, Korea; shparkmd@korea.ac.kr; 2Division of Foregut Surgery, Department of Surgery, Korea University Anam Hospital, Goryeodae-ro 73, Seongbuk-gu, Seoul 02841, Korea; 3Department of Surgery, Min General Surgery Hospital, 155 Dobong-ro, Gangbuk-gu, Seoul 01171, Korea; drlawyer@naver.com

**Keywords:** review, minimally invasive surgery, laparoscopy, robot surgery, navigation surgery, image-guided surgery, indocyanine green, bariatric surgery, metabolic

## Abstract

Since its introduction in the early 1990s, laparoscopic gastrectomy has been widely accepted for the treatment of gastric cancer worldwide. In the last decade, the Korean Laparoendoscopic Gastrointestinal Surgery Study group performed important clinical trials and exerted various efforts to enhance the quality of scientific knowledge and surgical techniques in the field of gastric cancer surgery. Laparoscopic gastrectomy has shifted to a new era in Korea due to recent advances and innovations in technology. Here, we discuss the recent updates of laparoscopic gastrectomy—namely, reduced-port, single-incision, robotic, image-guided, and oncometabolic surgery.

## 1. Introduction

Since its introduction by Japanese surgeons in the early 1990s, in the last 30 years, laparoscopic surgery has rapidly gained international acceptance for the treatment of gastric cancer (GC) [1,2]. The Korean Laparoendoscopic Gastrointestinal Surgery Study (KLASS)-01 trial demonstrated that laparoscopic distal gastrectomy provided patients with early gastric cancer (EGC), with better cosmetic effects and pain reduction than open surgery [3,4,5,6,7]. The KLASS 02 trial demonstrated that laparoscopic distal gastrectomy for patients with advanced gastric cancer (AGC) was associated with a lower complication rate, faster recovery, and less pain with equivalent oncologic outcomes than open surgery [8,9].

In the last decade, Korean laparoscopic surgeons have made consistent efforts to improve the technique of laparoscopic gastrectomy (LG). Indeed, LG has shifted to a new era in Korea due to the increase in surgeons’ experience and recent innovations in surgical instruments and technology. The clinical advantages of LG can be further enhanced by reducing the number of trocars or the length of the incision. Therefore, reduced-port laparoscopic gastrectomy (RPLG) and single-incision laparoscopic gastrectomy (SILG) are increasingly performed in Korea [10,11,12,13,14,15,16,17,18,19,20]. As the surgical robot system has the advantages of precise movement without tremor and articulation with more degrees of freedom, surgeons have tried to overcome unresolved issues of laparoscopic surgery by performing robotic gastrectomy (RG) [21,22,23,24,25]. In addition, fluorescent image-guided navigation technology has achieved satisfactory outcomes in nodal localization, function preservation, and oncologic quality improvement [26,27,28,29,30]. One of the most astonishing outcomes is that obese patients with GC and diabetes have improved glycemic control after surgery for GC. These findings resulted in the use of the terminology “oncometabolic surgery” by Korean surgeons [31].

In this review, we describe current updates and issues regarding LG from the perspectives of reduced-port, single-incision, robotic, image-guided, and oncometabolic surgery.

## 2. Reduced-Port and Single-Incision Laparoscopic Gastrectomy

### 2.1. Concept

RPLG was developed to overcome the technical difficulties of SILG. SILG, a surgery that integrates various efforts of minimal invasiveness by decreasing abdominal trauma, was first performed by Omori et al. [32] in 2011. This approach offers excellent cosmetic results because the scar is almost hidden in the umbilicus. Therefore, laparoscopic surgery should be performed via a single umbilical incision using a specially designed multichannel port. While conventional five-port LG requires triangulation regarding visualizing the laparoscopic surgical field and maneuvering the operator’s hands, SILG has a single dimension of surgical instruments and can be technically demanding even for experienced surgeons. Therefore, restriction of the working field and interference of laparoscopic instruments are the main technical issues associated with SILG [33]. However, additional ports and other lifting devices can alleviate these issues. Three-port totally laparoscopic distal gastrectomy (TLDG) uses one umbilical trocar for the laparoscopic camera and two trocars for the operator’s hands. Three-port TLDG is also called “duet-TLDG,” which emphasizes the fact that it is performed by a surgeon and scopist alone [16]. Two-port (dual-port) TLDG with an umbilical multichannel port plus one additional trocar is another option for overcoming the difficulties of SILG. RPLG does not require specialized instruments such as flexible scope or curved forceps; it can be easily performed by laparoscopic surgeons who are familiar with conventional LGs [14,19].

### 2.2. Operative Procedures

During reduced-port TLDG, the patient was placed in a reverse Trendelenburg supine position with the operator standing on the right side. Unlike some Japanese and Chinese surgeons who perform LG from the left side of the patient and then move to the right side, most Korean surgeons sat on the right side of the patient throughout the surgery. The scopist sat on the right side of, and caudal to, the patient. For R-duet TLDG, a 12 mm diameter trocar was inserted in the umbilical area (mainly for laparoscopy), and a 5 mm diameter trocar was placed in the right upper quadrant (RUQ) area, and 12 mm trocars in the right lower quadrant area [20]. For dual-port TLDG, a multichannel port (Gloveport, Nelis, Bucheon, Korea) was placed through the longitudinal 2.5–3 cm transumbilical incision. A 5 mm trocar was placed in the RUQ area [10].

During SILG, the patient was in a lithotomy supine position with reverse Trendelenburg. Meanwhile, the operator and scopist were positioned between the patient’s legs. A longitudinal 2.5 cm transumbilical skin incision was made. A commercial four-hole single port (Gloveport, Nelis) was then placed in the umbilical incision, and the abdominal cavity was insufflated with carbon dioxide at a pressure of 13 mmHg; no additional assistant trocar was used. A 10 mm flexible high-definition scope (Endoeye flexible HD camera system; Olympus Medical Systems Corp., Tokyo, Japan) and a 45 cm Harmonic scalpel (Ethicon Endo-Surgery Inc., Raritan, NJ, USA) were used to visualize every corner of the operative field and to facilitate dissection (Figure 1a) [11,12].

Laparoscopic lymph node dissection was meticulously performed with each surgeon’s unique and newly introduced methods [10,11,12,20] while complying with the Japanese gastric cancer treatment guidelines (Figure 1b) [34]. After transecting the proximal side of the stomach, the specimen was extracted via a multichannel port in dual-port TLDG, and a 2.5–3.0 cm extended incision was made at the umbilical trocar insertion site in R-duet TLDG. For SILG, the transected specimen was retrieved via a single umbilical incision without any extension.

During R-duet TLDG, Billroth II (B-II), uncut, or conventional Roux-en-Y (RY) reconstruction can be employed. Usually, the entry holes were made at both the greater curvature side of the stomach and jejunum approximately 15 cm distal from Treitz’s ligament. Then, each arm of the endoscopic linear stapler (ENDO REACH^®^; Reach Surgical Inc., TEDA, Tianjin, China) was inserted into the remnant stomach and jejunum, vertical anastomosis was made by firing the endoscopic linear stapler. The common entry hole was closed with one endoscopic linear stapler or a hand sewing suture, depending on the surgeon’s preferences. Jejunojejunostomy was made in a similar fashion. When performing a delta-shaped Billroth-I (B-I) reconstruction in a conventional five-port TLDG, the linear stapler should enter the left lower quadrant (LLQ) trocar for safe anastomosis [17]. Therefore, B-I reconstruction is not easy to perform during R-duet TLDG. However, the linear stapler approach is facilitated through the umbilical multichannel port in dual-port TLDG rather than using the RLQ port in R-duet TLDG [10].

### 2.3. Technical Feasibility and Surgical Outcomes

Many Korean studies have demonstrated the technical feasibility and comparable surgical outcomes of three-port TLDG compared with conventional LG [14,15,16,18,20]. In addition, some authors have reported that three-port (R-duet) TLDG also allows high-quality lymph node (LN) dissection in patients with EGC [14,16]. However, to date, in Korea, the only improved surgical outcomes that have been reported include the following: less operative pain and scarring, reduced medical costs, and requiring fewer assistants.

Regarding dual-port LG, enhanced minimal invasiveness has been reported in Japan and China. The dual-port TLDG was reported to have a lower postoperative complication rate than the conventional five-port TLDG [35]. Kawamura et al. reported that the amount of oral intake during the early postoperative period after dual-port TLDG exceeds that following conventional TLDG [36]. Shorter hospital stays and lower serum CRP levels were also demonstrated in dual-port TLDG [37]. Recently, a Korean single-arm study compared the surgical outcomes of dual port with those of dual-port TLDG and revealed that there were no significant differences in hospital stays, postoperative morbidities, operative time, time to first flatus, and diet between the two groups [10].

Korean and Japanese SILG case-matching studies analyzing the initial cases of pure single-port laparoscopic distal gastrectomy (SILDG) reported that it is both safe and feasible for patients with GC, with similar operation times and better short-term outcomes than conventional five-port TLDG including shorter hospital stays, earlier initiation of oral intake [38], less postoperative pain, less estimated blood loss, less inflammatory reaction, and improved cosmetic results [11,38]. In contrast to the absence of randomized controlled trials (RCTs) comparing SILDG with conventional or reduced port TLDG in Korea, a recent Japanese RCT demonstrated that SILDG is safe and feasible for the treatment of clinical stage I GC with better short-term results in terms of less severe pain and shorter operative time [39].

Regarding the surgical techniques of SILG, to date, only limited experiences for SILDG with B-II or RY anastomosis have been reported due to technical difficulties with B-I anastomosis [11,13], and the increased possibility of complications with esophagojejunostomy in laparoscopic total gastrectomy [40,41]. These anastomotic procedures are considered more difficult because advanced laparoscopic manipulations related to the linear stapler between the operator and assistant are essential to overcome the inherent poor ergonomics of SILG. However, a new generation of laparoscopic surgeons in Korea has recently attempted to overcome these difficulties. SILDG with unaided delta-shaped anastomosis [42], solo single-incision laparoscopic total gastrectomy (SILTG) using a laparoscopic scope holder instead of a scopist [12], and single-incision laparoscopic proximal gastrectomy with double tract reconstruction [43] were successfully performed with good reproducibility. In addition, surgeons utilized the benefits of technological advances in surgical instruments. Kang et al. showed the potential benefits of using a three-dimensional (3D) laparoscopic camera, including shorter operative time and lesser blood loss than two-dimensional cameras [44]. The possibility of increased freedom in maneuvering the laparoscopic grasper was demonstrated using a multi-degree-of-freedom articulating device with a 360° angle of motion [45].

### 2.4. Oncologic Validity

To date, the long-term oncologic validity of reduced-port TLDG has rarely been investigated. A Korean retrospective multi-institutional study analyzed 1117 patients with GC and revealed that three-port and conventional five-port TLDG groups showed no significant differences in 5-year overall survival (94.3% vs. 96.7%, *p* = 0.138) or disease-free survival (94.3% vs. 95.9%, *p* = 0.231) [46]. No randomized controlled trial has addressed this issue; therefore, we anticipate that the noninferiority of reduced port LG will soon be verified by clinical trials comparing the oncologic safety between reduced-port and conventional laparoscopic distal gastrectomy for EGC.

On the other hand, studies analyzing the survival rate of SILG are still lacking in Korea. A recent Japanese study with propensity score matching analysis revealed that the 5-year survival rates were 74.2% in the SILG group and 60.2% in the five-port conventional TLG group (*p* = 0.081 by log-rank test) [47].

### 2.5. Controversial Issues and Future Perspectives

The application of RPLG or SILG in GC treatment remains controversial, and some surgeons have critically commented on these issues. LG should be conducted by experts and should never fail to exert the best of their abilities. Due to the possibility of interference and collision of surgical instruments, the best performance can be achieved by placing an additional port when necessary [48]. Moreover, the fourth edition of the Japanese gastric cancer treatment guidelines states that LG for GC treatment is an option for patients with stage I disease [34]. The KLASS-01, 02 trials demonstrated comparable long-term oncologic outcomes of laparoscopy-assisted distal gastrectomy, compared with open distal gastrectomy [4,5,9]. However, long-term results of KLASS 06 trials analyzing the 3-year relapse-free survival rate of laparoscopic total gastrectomy, compared with those of open conventional surgery in patients with AGC, are awaited. It has been reported that achieving the learning curve of SILDG requires only 30 cases; however, this result was obtained due to the fact that surgeons with sufficient experience operated on patients with low BMI and a high female ratio.

Why do surgeons still perform RPLG or SILG? In recent decades, the shortage of surgical manpower has become a major problem, in Korea as well as in the USA and Europe [49,50]. Many medical college students and interns regard surgery as a dirty, dangerous, and difficult job, and applicants for surgical training are decreasing in number. Finally, surgeons must perform laparoscopic procedures alone or only with the scopists. Minimal invasiveness and elimination of the need for an assistant are beneficial. Here, we emphasize the educational issues and special training that junior and next-generation surgeons should consider.

Compared with open gastrectomy, laparoscopy-assisted gastrectomy (LAG) is associated with a significant reduction in respiratory complications [51,52]. A recent Korean retrospective study demonstrated that totally laparoscopic gastrectomy (TLG) provides a lower rate of pulmonary complications than LAG [53]. Upper abdominal incision, similar to LAG, and a larger sized incision is associated with worsened abdominal pain and decreased pulmonary function. Larger incisions in the epigastrium and significant pain can reduce movement of the diaphragm and deep breathing in the LAG [54,55]. Therefore, RPLG or SILG is expected to provide a much lower rate of pulmonary complications. However, studies with RPLG or SILG have rarely addressed pulmonary complications. Future randomized controlled trials investigating RPLG or SILG should also consider the impact of minimal invasiveness on pulmonary complications.

From an oncologic perspective, tumor localization is challenging owing to the invisibility of tumors on the serosal surface during TLG. Laparoscopic procedures deter surgeons from touching or sensing tumors or marking clips directly. Park et al. [56] demonstrated that intraoperative endoscopy can provide oncologic safety with no tumor cells in the resection margin during TLDG, avoiding unnecessary total gastrectomy. Considering that RPLG and SILG are totally laparoscopic procedures, the wider adoption of intraoperative endoscopy should also be considered during RPLG and SILG.

## 3. Robotic Gastrectomy

### 3.1. Concept

Although the clinical advantages of LG have been demonstrated in terms of less postoperative pain, shorter hospital stays, and faster gastrointestinal function recovery, some problems remain unresolved, such as D2 lymph node dissection, anastomosis technique, and oncologic safety in AGC [8,9]. The da Vinci^®^ Surgical System (Intuitive Surgical Inc., Sunnyvale, CA, USA) was introduced to overcome the drawbacks of laparoscopic surgery. Robotic surgical systems offer several advantages, including high-resolution 3D images, EndoWrist^®^ with seven degrees of freedom, and tremor filtering. Thus, this technique is expected to increase the accuracy and thoroughness of minimally invasive gastrectomy [24,25]. Since the introduction of RG in 2003 in Japan [57], Korean laparoscopic surgeons have adopted this new surgical approach to overcome the limitations of LG.

### 3.2. Technical Feasibility and Surgical Outcomes

Some previous studies in Korea reported that RG had less intraoperative estimated blood loss than conventional LG [23,25]. This was in line with the results of previous studies with retrospective and systemic meta-analysis designs [58,59]. Park et al. [60] reported that reduced blood loss was more apparent in specific subgroups, such as patients with AGC who underwent D2 LN dissection or patients with a high body mass index (BMI). In patients with high BMI and D2 LN dissection, estimated blood loss was significantly lower in the robotic distal gastrectomy group than in the conventional laparoscopic group [61]. This can be attributed to 3D imaging of the surgical field and the ability to finely dissect lymphatics from surrounding vessels with finer EndoWrist movement. While the reduction in blood loss provides better visibility of the surgical field, thus resulting in enhanced quality of the surgery, its impact on immediate postoperative outcomes is negligible. However, it may have oncologic benefits because it can minimize the dissemination of free cancer cells floating in blood and accumulation of blood, which creates a microenvironment suitable for tumor growth [62,63,64].

Due to the advantages of RG—namely, less intraoperative bleeding, surgeons expect RG to have better postoperative outcomes; however, there were no noticeable results in practice. In Korea, most studies, including retrospective and nonrandomized multicenter prospective designs, recently demonstrated that perioperative surgical outcomes of RG were similar to those of LG [21,22]. In contrast, a Japanese multi-institutional prospective study showed that RG in clinical-stage I/II GC provided a significantly lower complication rate than the historical comparison laparoscopic gastrectomy group (2.45% vs. 6.4%) [65]. In the future, a well-designed RCT investigating the surgical outcomes of RG, compared with those of conventional LG, is needed to provide high levels of evidence.

### 3.3. Operative Time and Learning Curve

A longer operative time has been reported in RG than in LG [58,59,66]. In Korea, a recent prospective multicenter study demonstrated that RG requires a longer operative time than LG [67]. Robotic surgery has inevitable time-consuming procedures, such as preoperative robot docking and preparation, changing of instruments performed by the assistant, and controlling the camera and instrument performed by the operator [68]. In addition, the surgeon that operates alone needs to perform the roles of assistant and camera scopist, which prolongs the operative time for RG [69]. Another factor that increases the operative time in RG is the initial learning curve phase of procedures [70]. A recent single institutional study in Korea reported that after completing the learning curve of 35 cases, the late group had a shorter console time (186.6 min vs. 247.1 min, *p* < 0.001) and operative time (281.7 min vs. 420.8 min, *p* < 0.001) than the early group before achieving the learning curve of 25 cases [71]. Likewise, operative time and docking time were significantly reduced after a learning curve of 25 cases in another Chinese retrospective study [72]. If the inevitable time-consuming procedures during RG were easier to perform, the operative time for RG would be similar to that of laparoscopic surgery. When performed by an experienced surgeon, the operative time in RG was not longer than in LG [21]. In the future, technological innovations, such as the autofocusing ability of the robotic camera, multifunctional instruments, and a rapid instrument changing system, are expected to further reduce the operative time of RG compared to current practice.

Generally, surgeons are required to perform 20–25 cases to overcome the learning curve for RG, which is considerably less than the 60–90 cases required for LG [21,71,72,73]. Due to similarities in the operative procedure for RG and LG, performing sufficient laparoscopic procedures enable surgeons to easily perform RG. Park et al. [74] reported that an experienced laparoscopic surgeon required only 10 cases to adapt to RG. Indeed, a Korean surgeon with extensive experience, specifically 6000 open gastrectomy procedures, but with limited experience in laparoscopic procedures, also successfully achieved stabilization in the operative time of RG after completing 25 cases [71]. Unlike previous studies that analyzed the learning curve focusing on operative time, a recent Korean study demonstrated that the critical learning curve period itself can affect the patient’s postoperative courses by analyzing the learning curve based on the postoperative complication rate [75].

### 3.4. Oncologic Outcomes

Retrieving an adequate number of LN can offer accurate cancer staging with a more precise LN ratio system, thus improving survival in patients with GC [76,77]. LG with D2 LN dissection for AGC is a technically demanding procedure and is associated with high morbidity and mortality rates. Therefore, LG is typically considered a treatment option for patients with EGC. On the other hand, articulation with seven degrees of freedom and a comfortable environment with tremor elimination in robotic systems can enable surgeons to meticulously discern and finely dissect lymph nodes from surrounding complex lymphovascular structures or vital organs. Thus, RG is expected to be helpful in technically challenging procedures [25]. LN dissection near the supra-pancreatic or splenic-hilar area is considered one of the most technically difficult procedures in LG. Indeed, recent comparative studies showed that RG harvested more LNs from the suprapancreatic area than laparoscopic surgery [78]. Moreover, the number of harvested LNs at LN stations 10 and 11 was significantly higher for RG than for laparoscopic surgery [24].

Several studies have adopted propensity-score matching analysis and investigated the long-term outcomes of RG for GC. In Korea, a large-scale, retrospective, single-institutional cohort study demonstrated that robotic and laparoscopic gastrectomy have similar 5-year overall survival rates (OS) (93.2% vs. 94.2%, *p* = 0.521) and relapse-free survival (RFS) (90.7% vs. 92.6%, *p* = 0.229), respectively [79]. Roh et al. [80] also reported that there were no differences in the 3-year OS (98.6% and 89.7%, respectively; log-rank *p* = 0.144) and RFS between the RTG and LTG groups (97.3% and 87.0%, respectively; log-rank *p* = 0.167). A recent study investigating the 14-year experience of 2000 RG procedures demonstrated that the 5-year OS rates were 97.6% for stage I, 91.9% for stage II, and 69.2% for stage III, with a total recurrence rate of 5.3% [81]. These results are comparable with those of previous studies comparing the long-term outcomes of RG procedures with those of laparoscopic surgeries [82,83,84]. These results indicate that RG for GC treatment may be safer than laparoscopic surgery. However, an insufficient follow-up period, a high proportion of early stage disease, and a lack of well-designed randomized controlled trials contribute to the inability to confirm the safety of RG. Although there are practical constraints, RCT is ultimately necessary to determine the oncologic safety of RG.

### 3.5. Future Perspectives

Recently, reduced port and single-incision surgery have been introduced in GC surgery, showing acceptable and feasible outcomes when performed by experienced surgeons [10,11,12,13,19,20,43,46]. These trends are also reflected in RG. In Korea, a phase I/II clinical trial demonstrated the safety and feasibility of reduced-port RG for EGC [85]. Seo et al. [86] introduced a modified technique using an infraumbilical single-site and two additional ports in the field of reduced-port totally robotic distal gastrectomy (Figure 2a,b). This novel technique resulted in acceptable surgical outcomes, with blood loss of 49.9 mL, operative time of 210 min, and the mean number of retrieved LNs of 58.8. The delta-shaped Billroth I reconstruction technique was applicable during reduced-port totally robotic distal gastrectomy using a similar approach with an infraumbilical single-site and two additional ports [87]. To date, there is no level I evidence that RG can further expand into minimally invasive gastrectomy. Considering the advantages of robotic surgical systems, RG is expected to overcome technically challenging procedures in the future.

## 4. Fluorescence Image-Guided Gastrectomy

### 4.1. Concept

Resection of a sufficient number of lymph nodes (LNs) has become the current requirement for improved survival during GC surgery. In addition, it provides accurate staging using a precise LN ratio system [76,88]. However, radical laparoscopic lymphatic dissection is a technically demanding procedure due to complex vasculatures with multiple lymphatic channels surrounding the stomach. Careless handling and manipulation of the lymphovascular structures carry a high risk of tissue or vascular injury, which can lead to intra- or postoperative complications, such as fatal bleeding and pancreatic fistula [89,90,91]; however, it remains a substantial challenge, even for surgeons with a high level of proficiency. On the other hand, favorable and excellent outcomes obtained after treatment of patients with EGC have enabled the development of function-preserving gastrectomy with a focus on postoperative quality of life (QoL). Sentinel node (SN) negativity following frozen section analysis is essential for function-preserving gastrectomy [92]. Therefore, novel techniques to alleviate these obstacles during LG are required.

Image-guided surgery has been introduced in the field of surgical oncology due to recent advances in surgical technology. Indocyanine green (ICG) is a water-soluble tricarbocyanine dye that binds to albumin and has lower toxicity. Near-infrared (NIR) light has longer wavelengths and allows greater penetration into tissues. ICG-guided NIR imaging can provide better visualization of lymph nodes and lymphatic channels than other dying materials under visible light. Fluorescent image-guided surgery can help surgeons obtain additional anatomical information, such as identifying lymph nodes in thick fatty tissues, discerning lymphatic channels from vasculatures, identifying the origin and shape of a vessel, and assessing tissue perfusion status [26,93,94,95,96,97].

### 4.2. Oncologic Outcomes

As a new surgical navigation technique, ICG-guided NIR fluorescent imaging has shown improved oncologic quality in laparoscopic or robotic GC surgery. A recent prospective RCT demonstrated that the ICG NIR tracer-guided LG group had a noticeably increased number of dissected LNs and reduced LN noncompliance, compared with the conventional LG group (49.6 ± 15.0 vs. 41.7 ± 10.2%, *p* < 0.001; 31.8% vs. 57.4%, *p* < 0.001) [98]. In Korea, a prospective single-arm study revealed that the number of retrieved LNs was larger in the NIR-guided RG group than in the historical controls with RG (48.9 ± 14.6 vs. 35.2 ± 11.2; *p* < 0.001), and the use of ICG was particularly useful for the identification and dissection of the LNs in the infrapyloric and suprapancreatic regions [28]. Kim et al. [27] showed that the clinical application of a NIR imaging can provide additional node detection, resulting in complete LN dissection in the infrapyloric region, which is a technically challenging area.

### 4.3. Sentinel-Node Navigation Surgery

The application of intraoperative SN biopsy is expected to reduce unnecessary radical lymphadenectomy and allow function-preserving gastrectomy. In Korea, a prospective phase II trial was conducted to confirm the safety and feasibility of SN-navigation LG. Patients with positive SNs underwent gastrectomy with radical lymph node dissection (SN-positive group), whereas those with negative SNs received only limited gastric resections without further lymphadenectomy. The 3-year OS and RFS rates for the SN-negative group were 97.7% (95% C: 94.7–100.0%) and 95.5% (95% CI: 91.3–99.9%), respectively [99]. The SENORITA group conducted a prospective multicenter phase III trial to determine the oncologic safety of limited gastric resections with sentinel basin dissection (SBD), compared with conventional LG [29]. An interim analysis reported that the 3-year DFS rate following limited gastric resection with SBD was not different from that after conventional LG (93% vs. 96%). We expect the long-term results of this trial to clarify issues regarding the oncologic safety of SN-navigation LG [100].

### 4.4. Technical Advantages

Based on a technical viewpoint, Park et al. [30] demonstrated that NIR fluorescence guidance can provide safe and fast infrapyloric lymphadenectomy in laparoscopic distal gastrectomy. The operative time for infrapyloric LN dissection was significantly shorter in the ICG group than in the non-ICG group (13.1 ± 5.8 vs. 18.7 ± 7.9 min; *p* = 0.001), and the incidence of bleeding during infrapyloric LN dissection was lower in the ICG group (20% vs. 68.3%, *p* < 0.001). Identification of the infrapyloric artery (IPA) type is essential for safe pylorus-preserving gastrectomy. By visualizing the blood vessels and flow vividly, real-time NIR fluorescence navigation identified the IPA type, with a prediction rate of 80% [26]. A single-arm study also demonstrated that NIR fluorescence imaging can facilitate the identification of aberrant left hepatic arterial (ALHA) territories. After clamping the ALHA, ICG-guided fluorescence was used to visualize the liver using a NIR camera. This novel and simple technique helped surgeons decide whether to preserve or ligate an ALHA [101]. Insufficient blood supply is an important risk factor for anastomosis site leakage during gastrointestinal surgery [102,103]. Huh et al. [104] demonstrated that intraoperative ICG angiography using a NIR camera can successfully assess vascular perfusion status at the anastomosis site [101].

### 4.5. Controversial Issues and Future Perspectives

Better visualization of lymphatic channels, vessels, and LNs during GC surgery can encourage surgeons to achieve completeness of the lymphatic dissection without breakage of lymphatic structures. These detailed efforts can prevent tumor cell spillage and dissemination, ultimately resulting in improved oncologic quality. Moreover, complex, multiple lymphovascular structures within the stomach can become obstacles even for experienced surgeons. Fluorescence image guidance can help surgeons perform safer and even faster lymphatic dissection by preventing unexpected injuries when dissecting lymph node–bearing fatty tissue around blood vessels (Figure 3a,b).

Compared with previous modalities, ICG-guided NIR imaging can detect even a very small amount of ICG and show too many LNs from the laparoscopic surgical view, which may be a source of confusion in clinical applications [105,106]. Quantification of the ICG signal may provide more precise information to discriminate each LN-bearing tissue with different levels of fluorescence uptake. To date, the protocol guideline for the ICG technique has not yet been established. For example, Kwon et al. performed a peritumoral injection of ICG of 1.25 mg/mL solution 1 d before surgery. Additionally, Kwon et al. opted for endoscopic peritumoral injection of ICG (1.25 mg/mL) administered 1 d before RG to allow sufficient distribution of fluorescent ICG. Intraoperative endoscopy can prolong operative time and disrupt the laparoscopic surgical view by insufflating air into the small intestine. However, detection of the first LN with ICG uptake by the NIR camera is relatively quick (approximately 3 min). The ICG fluorescence signal is typically washed out 1 h after injection and almost disappears 1 d later [107]. Therefore, preoperative ICG injections should be reconsidered. By applying a laparoscopic intestinal bulldog clamp to the jejunum 10–15 cm below the Treitz ligament, recent single-arm studies assessed the value of intraoperative submucosal injection of ICG (0.1 mg or 0.5 mg/mL), at four or five different anatomical locations [27,30]. One study reported that 80 to 90 cases were required to overcome the learning curve of endoscopic procedures [108]. When surgeons have sufficient experience with intraoperative endoscopy, an intraoperative injection can be performed within 5 min [30]. Future RCTs are expected to provide stronger evidence for optimal timing and concentration of ICG injection.

## 5. Oncometabolic Surgery

### 5.1. Concept

Originally, bariatric surgery was reported to cure morbid obesity; however, it was also highly effective in the treatment of chronic comorbidities of obese patients, such as type 2 diabetes mellitus (T2DM), dyslipidemia, and hypertension [109]. Among these, improvement of T2DM at 2–5 years postoperative has been particularly excellent, specifically 48.9–75.2% [110,111,112,113]. Bariatric surgery induced glycemic control independent of the resultant weight loss. These findings gave rise to the concept of “metabolic surgery” [13].

GC surgery and bariatric/metabolic surgery have similar operative procedures, including gastric resection and foregut bypass. Therefore, it can be hypothesized that GC surgery also has beneficial effects on patients’ glycemic control. Indeed, the improvement rate of T2DM after GC surgery was similar to that after bariatric/metabolic surgery [114,115,116,117]. These findings inspired the emergence of the terminology “oncometabolic surgery” [31], which targets the removal of malignancy and improved glycemic control with a one-step procedure. Considering that the incidence of T2DM is gradually increasing and that it is associated with increased mortality of patients with GC, oncometabolic surgery is expected to improve QoL and prolong the survival of patients with GC. However, a comparison of the baseline properties and operative procedures of the GC patient population and the obese patient population showed that these populations were not similar. Although the benefits of conventional GC surgery have already been confirmed regarding glycemic control, the degree of improvement may differ based on different operative procedures. Therefore, the procedures of oncometabolic surgery can be modified, carefully assessed, and engineered to maximize clinical benefit without compromising oncologic safety.

### 5.2. Patient Selection

Compared with bariatric populations, the patients who are candidates for oncometabolic surgery are typically older, have lower BMI, and present different pathophysiology for T2DM. The most notable difference between bariatric and oncometabolic populations is weight, and the degree of weight loss is an important contributing factor for improved glycemic control in bariatric surgery. Lee et al. [118,119] included the preoperative BMI in the ABCD score system, which aimed to predict the likelihood of improvement in glycemic control after metabolic surgery. Oncometabolic surgery should be reconsidered for patients with GC with a lower BMI who have difficulty achieving metabolic benefits from weight loss. Several Korean studies analyzing nonobese patients with GC investigated the impact of GC surgery on diabetes remission and demonstrated that surgery improved glycemic control in these patients [18,36,39]. However, other studies confirmed that preoperative BMI or perioperative changes in BMI were related to the improvement of glycemic control, suggesting that metabolic surgery cannot provide the benefit of glycemic control in patients with GC with lower BMI [19,36,39].

As patients with GC are older than patients in bariatric populations, their pathophysiologic mechanism for T2DM is also different. While older patients with GC are more likely to have pancreatic β-cell dysfunction, young obese patients tend to have decreased insulin sensitivity in peripheral tissue and impaired hepatic glucose metabolism. Lee et al. [118] reported that age is a negative predictive factor for the improvement of glycemic control in the ABCD and DiaRem scoring systems. Moreover, the surgery itself can be a risk factor for increased mortality in older patients. Therefore, the surgeon should also consider the patient’s age.

On the other hand, the “severity of T2DM” also affects the level of improvement in glycemic control. In Korea, several studies have demonstrated that patients with T2DM for a longer duration, that require insulin therapy, or present a higher level of preoperative HbA1c, do not typically benefit from surgery [31,115,120,121,122,123]. These findings indicate that patients with severe T2DM are less likely to experience improvement in glycemic control after metabolic surgery. Therefore, surgeons should carefully consider whether patients can benefit from glycemic control through oncometabolic surgery.

From an oncologic perspective, one recent Korean study suggested a possible treatment algorithm for patients with EGC with T2DM. Conventionally, patients with GC characterized by clinical T1aN0M0, ≤2 cm in size, differentiated, and without ulceration are recommended to undergo endoscopic submucosal dissection [34,124]. However, when these patients have T2DM, they require improved glycemic control for better QoL. For these patients, only ESD was effective, and radical resection or lymphatic dissection was not needed. Therefore, alternative treatment options, such as oncometabolic surgery, can be used to treat T2DM. In patients with GC in the greater curvature, sleeve gastrectomy, or TG, distal gastrectomy with long-limb RY reconstruction can be considered without compromising oncologic safety. Regarding tumors located in the lesser curvature, TG and distal gastrectomy with long-limb RY reconstruction can be recommended. This algorithm may bring about changes in the conventional guidelines that are currently in use.

### 5.3. Efficacy

In Korea, consistent efforts have been made to investigate the efficacy of GC surgery in the improvement of T2DM. However, studies reporting the efficacy of cancer surgery for improvement of glycemic status employed their own criteria for diabetes improvement (e.g., a decrease in the number of diabetic medications or a decrease in fasting plasma glucose or HbA1c levels). When the remission rates were displayed based on the American Diabetes Association (ADA) definition [125], the rate of diabetes remission ranged from 1.0% to 72.8% in partial gastrectomy with BII reconstruction group and from 27.3% to 90.5% in the TG with RY reconstruction group. Different follow-up periods and study designs may have resulted in wide variability in remission rates. Regardless of the employed criterion, TG with RY reconstruction is associated with the best efficacy in the improvement of glycemic control [31,115,119,120,122,126]. Generally, procedures with duodenal bypass (RY and BII reconstruction) showed better glycemic control than those without duodenal bypass (BI reconstruction) [115,119,121,122]. Weight-loss-induced metabolic effect on glycemic control may be minimal in nonobese patients with GC because patients with GC have a lower level of preoperative BMI than bariatric populations. Interestingly, a meta-analysis revealed that BII reconstruction was more beneficial than BI reconstruction in glycemic control for patients with less reduction in BMI [116].

In addition to conventional GC surgery, Korean surgeons have attempted to adopt modified operative techniques with longer alimentary and biliopancreatic limbs to mimic the maximization of nutrient malabsorption, similar to bariatric Roux-en Y gastric bypass procedures [120,127,128]. This new type of procedure is called long-limb RY reconstruction. A preliminary prospective study of 30 patients with long-limb RY reconstruction showed a 30% diabetes remission rate based on ADA definition and further general improvement in 20% of patients [120]. Recently, a retrospective study analyzing 226 patients demonstrated that long-limb RY reconstruction was superior to Billroth II in glycemic control in the one-year postoperative period [128]. However, the results were not obtained by comparing patients who underwent long-limb RY reconstruction with those who underwent conventional RY reconstruction, and it is difficult to determine whether the excellent glycemic control associated with long-limb RY reconstruction is due to increased limb length or RY reconstruction itself. Although diabetes remission and improvement were excellent in obese patients with GC, it was lower than that in bariatric populations. However, the benefit from metabolic surgery may not be higher in the patient population with GC than in bariatric populations owing to their lower BMI, older age, and less pancreatic β-cell function.

Long-limb RY reconstruction results in an enhanced degree of malabsorption; therefore, nutritional concern is a new issue in patients undergoing oncometabolic surgery. A pilot study analyzing 20 patients demonstrated that the cumulative incidences of anemia, iron deficiency, and vitamin B12 deficiency following long-limb RY reconstruction were not different from those after conventional RY reconstruction. However, the median vitamin B12 levels tended to be lower, and the reduction in vitamin B12 concentration tended to be higher in the long-limb RY reconstruction group [129]. These findings suggest that vitamin B12 levels should be carefully monitored after oncometabolic surgery with long-limb RY reconstruction.

### 5.4. Future Perspectives

T2DM is an important risk factor for mortality in patients with GC [130]. Improvement of diabetes following GC surgery is related to an increased 5-year survival rate [117]. These findings provide robust evidence that oncometabolic surgery may provide better survival outcomes in patients with GC with T2DM. Based on these findings, future studies are needed to investigate the impact of oncometabolic surgery on the survival of patients with GC.

## 6. Conclusions

GC surgery is still evolving through advances in surgical technology and devices, as well as the accumulation of knowledge and inspiration from older generations. However, given that long-term results of the described contents have not yet been verified, the current and updated surgical techniques and procedures should be carefully considered. Additionally, various efforts should be made to solve the issues of feasibility, training, education, and oncologic validity. Future prospective, well-designed multicenter studies are needed to provide reasonable and robust evidence for the current updated contents of GC surgery in Korea.

## Figures and Tables

**Figure 1 medicina-57-01195-f001:**
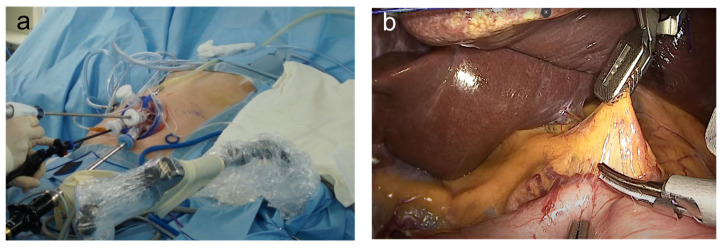
The current issues of single-incision laparoscopic gastrectomy in Korea: (**a**) outside view of single-incision laparoscopic gastrectomy; (**b**) laparoscopic view of single-incision laparoscopic gastrectomy.

**Figure 2 medicina-57-01195-f002:**
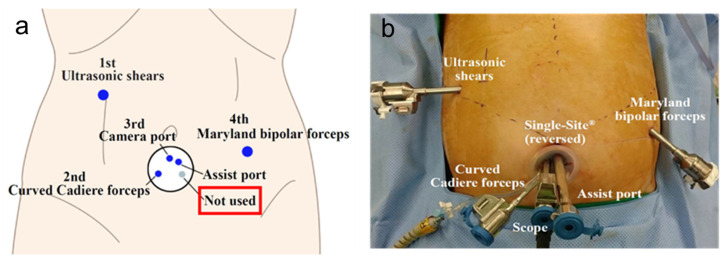
The current issues of robotic gastrectomy in Korea: (**a**) schematic illustrations of port placement of the single-site and two additional ports in robotic distal gastrectomy; (**b**) outside view of port placement of the single-site and two additional ports in robotic distal gastrectomy.

**Figure 3 medicina-57-01195-f003:**
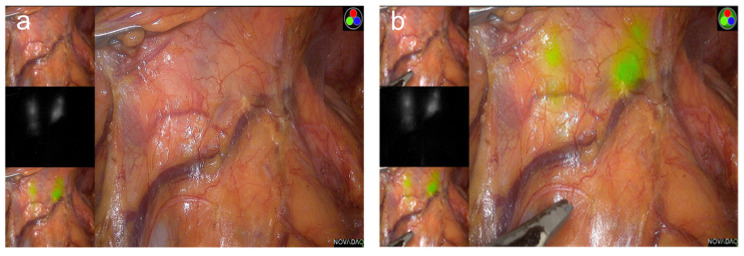
The current issues of fluorescence image-guided gastrectomy in Korea: (**a**) laparoscopic view under visible light during laparoscopic distal gastrectomy; (**b**) indocyanine green-enhanced fluorescence uptake of lymphatic channels and lymph nodes during laparoscopic distal gastrectomy.

## Data Availability

Not applicable.

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
