# Peer review of "Current Status and Trends of Minimally Invasive Gastrectomy in Korea"

_medicina, 2021, doi:10.3390/medicina57111195_

Round 1

Reviewer 1 Report

The Authors proposed a well conducted review on laparoscopic gastrectomy in Korea.

However, I think that this manuscript could be responding to minor concerns.

Title: I think that could be better to substitute the word "laparoscopic" with "minimally invasive" (there is a paragraph on robotic approach)

Results: I think that sould be useful to assess how the korean surgeon perform the anastomosis after gastrectomy (with stapler/manually) and how to close enterotomy after BII or RenY anastomosis.

Spelling mistakes are present througout the text.

Author Response

Comment #1) Title: I think that could be better to substitute the word "laparoscopic" with "minimally invasive" (there is a paragraph on robotic approach)

Answer #1) We revised the title according to reviewer's recommendations as follows in Page 1, Line 2.

: Current status and trends of laparoscopic minimally invasive gastrectomy in Korea.

Comment #2) Results: I think that should be useful to assess how the Korean surgeon perform the anastomosis after gastrectomy (with stapler/manually) and how to close enterotomy after BII or Roux-en Y anastomosis.

Answer #2) We revised and added the anastomotic procedures after gastrectomy according to reviewer's recommendations as follows in Page 3, Line 103-109.

: During R-duet TLDG, Billroth II (B-II), uncut, or conventional Roux-en-Y (RY) reconstruction can be employed. Usually, the entry holes were made at both the greater curvature side of stomach and jejunum approximately 15 cm distal from Treitz’s ligament. Then, each arm of the endoscopic linear stapler (ENDO REACH®; Reach Surgical Inc., TEDA, Tianjin) was inserted into the remnant stomach and jejunum, vertical anastomosis was made by firing the endoscopic linear stapler. The common entry hole was closed with one endoscopic linear stapler or a hand sewing suture, depending on surgeon’s preferences. Jejunojejunostomy was made by similar fashion. When performing a delta-shaped Billroth I (B-I) reconstruction in a conventional five-port TLDG, the linear stapler should enter the left lower quadrant (LLQ) trocar for safe anastomosis [17]. Therefore, B-I reconstruction is not easy to perform during R-duet TLDG. However, the linear stapler approach is facilitated through the umbilical multichannel port in dual-port TLDG rather than using the RLQ port in R-duet TLDG.

Comment #3) Spelling mistakes are present throughout the text.

Answer #3) Words such as “scopist”, “Billroth I”, “EndoWrist®”, “lymphovascular”, “suprapancreatic”, “Infrapyloric”, “tricarbocyanine”, and “oncometabolic” were automatically displayed with red lines throughout the manuscript. However, these terminologies are not spelling mistakes or ungrammatical expressions.

The scopist means who holds and operates the laparoscopic camera. This word was appeared in many articles related with laparoscopic gastrectomy.

Billroth I and tricarbocyanine is a proper noun.

EndoWrist refers to the brand name.

Lymphovascular refers to a condition in which lymphatics and vascular structures are intertwined and is a frequently mentioned expression in gastric cancer surgery-related papers.

Suprapancreatic refers to a location above the pancreas, infrapyloric refers to a location below the pylorus, respectively. These terms are surgical terminology, which were frequently mentioned in gastric cancer surgery-related papers.

As we have already mentioned in lines 444-445, the word oncometabolic is a new terminology that appeared in the [31] journal, meaning that cancer surgery has both cancer removal and metabolic effects.

However, we changed sentences as follows in Page 6, line 281 – 283.

Retrieving an sufficientadequate number of LN dissections can offer accurate cancer staging with a more precise LN ratio system, thus improving survival in patients with GC [76, 77].

The word “long-lib RY reconstruction” was changed to “long-limb RY reconstruction” in Page 11, line 520.

Author Query #1) We included one more author, who made considerable contribution, with interpretation of articles, critical revision of manuscript, as follows in Page 1, line 9.

3Department of Surgery, Min General Surgery Hospital, 155 Dobong-ro, Gangbuk-gu, Seoul 01171, Seoul, Republic of Korea.; [email protected]

Reviewer 2 Report

This is a fine narrative summary of progress in laparoscopic gastrectomy that will be of interest to international UGI surgeons. It is clear, well written, and free of obscurities. The authors are to be congratulated.

Author Response

I really appreciate the reviewer's valuable comment.

Reviewer 3 Report

Interesting review paper.

Author Response

(The authors gave the same response as above.)

Reviewer 4 Report

it is a thorough and well structured review the text is clear and the figures are representative

Author Response

(The authors gave the same response as above.)
